# RETRACTED: Variability in Enteric Methane Emissions among Dairy Cows during Lactation

**DOI:** 10.3390/ani13010157

**Published:** 2022-12-31

**Authors:** Ali Hardan, Philip C. Garnsworthy, Matt J. Bell

**Affiliations:** 1School of Biosciences, The University of Nottingham, Sutton Bonington Campus, Loughborough LE12 5RD, UK; phil.garnsworthy@nottingham.ac.uk; 2Animal and Agriculture Department, Hartpury University, Gloucester GL19 3BE, UK; matt.bell@hartpury.ac.uk

**Keywords:** cattle, methane, measurements, farm

## Abstract

**Simple Summary:**

The objective of this study was to investigate variability in enteric methane (CH_4_) emission rate and emissions per unit of milk among dairy cows on commercial farms in the UK. A large dataset of enteric CH_4_ measurements from individual cows was obtained from 18 farms across the UK. We conclude that changes in CH_4_ emissions appear to occur across and within lactations, but ranking of a herd remains consistent, which is useful for obtaining CH_4_ spot measurements.

**Abstract:**

The aim of this study was to investigate variability in enteric CH_4_ emission rate and emissions per unit of milk across lactations among dairy cows on commercial farms in the UK. A total of 105,701 CH_4_ spot measurements were obtained from 2206 mostly Holstein-Friesian cows on 18 dairy farms using robotic milking stations. Eleven farms fed a partial mixed ration (PMR) and 7 farms fed a PMR with grazing. Methane concentrations (ppm) were measured using an infrared CH_4_ analyser at 1s intervals in breath samples taken during milking. Signal processing was used to detect CH_4_ eructation peaks, with maximum peak amplitude being used to derive CH_4_ emission rate (g/min) during each milking. A multiple-experiment meta-analysis model was used to assess effects of farm, week of lactation, parity, diet, and dry matter intake (DMI) on average CH_4_ emissions (expressed in g/min and g/kg milk) per individual cow. Estimated mean enteric CH_4_ emissions across the 18 farms was 0.38 (s.e. 0.01) g/min, ranging from 0.2 to 0.6 g/min, and 25.6 (s.e. 0.5) g/kg milk, ranging from 15 to 42 g/kg milk. Estimated dry matter intake was positively correlated with emission rate, which was higher in grazing cows, and negatively correlated with emissions per kg milk and was most significant in PMR-fed cows. Mean CH_4_ emission rate increased over the first 9 weeks of lactation and then was steady until week 70. Older cows were associated with lower emissions per minute and per kg milk. Rank correlation for CH_4_ emissions among weeks of lactation was generally high. We conclude that CH_4_ emissions appear to change across and within lactations, but ranking of a herd remains consistent, which is useful for obtaining CH_4_ spot measurements.

## 1. Introduction

In terms of sustainability management in ruminant production, low-productivity systems lose more energy per unit of animal product than high-productivity systems [1]. Global milk production, and the number of milking cows, has increased in recent decades to meet increasing demand for dairy products, and with this, monitoring and mitigation of greenhouse gas emissions associated with milk production has gained importance [2]. Herd fertility, disease incidence and replacement rate are major influencers of CH_4_ emissions per kg of product [3,4]. Greenhouse gas emissions per unit product from dairy cows has been reducing by about 1% per annum over the last few decades with improved efficiencies of production [5]. However, due to increasing production per animal over this period, emissions per cow are estimated to have increased by 1.0% per annum [5].

Cattle are a notable source of CH_4_ emissions from fermentation of food consumed, with enteric emissions accounting for approximately 50% of total greenhouse gas emissions from milk production [2]. Genetic selection on CH_4_ could potentially help to mitigate emissions per cow and per unit product. A breath sampling or sniffer [4,6,7] approach to measure enteric CH_4_ emissions from individual cows on commercial farms now provides the opportunity to explore differences among farms and populations of animals in their normal environment. Respiration chambers are the gold standard method to obtain precise and accurate measurements of enteric CH_4_ emissions from individual animals. However, use of a mobile gas analyser approach to measure CH_4_ emissions has been found to be correlated with respiration chamber measurements, and provides a cheaper and wider scope of application to measure large numbers of animals in their normal environment [8].

Frequent ‘spot’ measurements of CH_4_ over several days from the same animal whilst being milked, along with application of signal processing to detect maximum amplitude of eructation peaks, has been found to provide a reliable and repeatable measure from individual animals using a mobile gas analyser [9,10]. Peaks in signal of CH_4_ concentration are caused by eructations when sampling emissions in breath of cows. Few studies have investigated phenotypic variation in CH_4_ emissions measured across commercial farms [6,7,11]. Such information is invaluable for quantifying normal variation and differences among systems with the potential for mitigating CH_4_ emissions.

The objective of the current study was to investigate variability in enteric CH_4_ emission rate and emissions per unit milk across lactations among dairy cows on commercial farms in the UK.

## 2. Materials and Methods

Approval for this study was obtained from the Animal Welfare and Ethical Review Board of the University of Nottingham before the commencement of the study (approval number 30/3210).

### 2.1. Data

A total of 105,701 CH_4_ spot measurements were obtained from 2206 dairy cows on 18 commercial farms using robotic milking stations in the UK (Table 1). Measurements were obtained between December 2009 and December 2013. Most cows in this study were Holstein-Friesian breed.

Cows were fed ad libitum and diets fed (Table 2) were classified as either a partial mixed ration (PMR; i.e., conserved forage and concentrate feed) or a PMR with grazed pasture. Pasture was predominantly perennial ryegrass swards. All cows received concentrate feed during milking allocated according to each cow’s daily milk yield. Milk yield, live weight and intake of robot concentrate were recorded automatically at every milking. Dry matter intake (DMI) of cows was estimated from individual milk yield and live weight using the equation DMI (kg/day) = 0.025 × live weight (kg) + 0.1 × milk yield (kg/day) [12]. Eleven farms fed PMR and 7 farms fed a PMR with grazing (farms A to R). Cows at farms B, C and E were sampled during two separate periods when the cows were fed either a PMR or PMR with grazing. Percentage of forage and concentrate in each diet were obtained, and all nutrient analysis was conducted by a commercial analytical laboratory (Sciantec Analytical Services, Cawood, UK). (Table 2). Farms all fed a commercial concentrate blend.

### 2.2. Gas Sampling

Methane concentration (parts per million, *v/v*) in breath samples collected during milking was measured using an infrared gas analyser (Guardian SP; Edinburgh Instruments Ltd., Livingston, UK) and recorded at 1 s intervals using a data logger (Simex SRD-99; Simex Sp. z o.o., Gdańsk, Poland). Breath samples were collected via a tube positioned at the rear of the feed bin in each robotic milking station, and sampling was carried out for at least 7 days at each farm. Raw logger data for CH_4_ concentration were analysed using MatLab Signal Processing Toolbox (version R2018a, The MathWorks, Inc., Natick, MA, USA). Peak analysis tools were used to identify eructation peaks and extract maximum amplitude within each milking. See Garnsworthy et al. [4] and Hardan et al. [10] for a full description of the sampling approach used.

The analyser sampled air at a flow rate of 1 L/min and measured CH_4_ concentration in ppm every second. Measurements were converted to emission rate in grams per minute by multiplying by 60 and density of CH_4_, assumed to be 0.706 × 10^−6^ g/L. Emission rates (grams per minute) were scaled to estimate emissions based on exponential rise time of eructation peaks and response time of the analyser (60 s) using Equation (1):CH_4_ emission rate (g/min) = maximum peak amplitude (ppm)/[1 − EXP (−(peak rise for amplitude in seconds/60))] × 60 × 0.706 × 10^−6^(1)

Emissions per kg milk were calculated by Equation (2).
CH_4_ emission rate (g/kg milk) = (CH_4_ emission rate (g/min) × 1440)/milk yield (kg/day)(2)

### 2.3. Statistical Analysis

Average values for week of lactation were used for analysis. A multiple experiment meta-analysis model in Genstat Version 21.1 (VSN International, Hemel Hempstead, UK) was used to assess effects of farm, week of lactation, parity, diet and dry matter intake on average CH_4_ emissions per individual cow using Equation (3):y_ijklm_ = µ + F_i_ + W_j_ + P_k_ + D_l_
*× β*DMI + C_m_ + E_ijklm_(3)
where y_ijklm_ is the dependent variable; µ is the overall mean; F_i_ is the fixed effect of farm (A to R); W_j_ = fixed effect of week of lactation (1 to 70); P_k_ = fixed effect of parity (1, 2, 3, 4 and 5 or more); D_l_ = fixed effect of diet (PMR or PMR with grazing); *β*DMI is the linear regression of Y on dry matter intake; C_m_ is random effect of individual cow; E_ijklm_ = residual error term.

Repeatability of gas concentration measures was assessed by σ^2^ animal/(σ^2^ animal + σ^2^ residual), where σ^2^ is the variance. Residual coefficients of variation (CV) were calculated from variance components as square root of residual σ^2^ divided by estimated mean. Spearman’s rank correlation was used to assess ranking of CH_4_ emissions from cows across farms and weeks of lactation. Significance was declared at *p* < 0.05.

## 3. Results

Across the 18 farms studied, cows averaged 2.6 ± 1.9 lactations, were milked 2.4 ± 0.8 times per day at robotic milking stations with CH_4_ measurements, and produced 30.4 ± 9.7 kg/day of milk (mean ± s.d.; Table 1). Estimated mean enteric CH_4_ emissions across the 18 farms were 0.38 (s.e. 0.01) g/min and 25.6 (s.e. 0.5) g/kg milk.

Factors with significant effects on CH_4_ emission rate were: farm, week of lactation, parity, dry matter intake and the interaction between dry matter intake and diet (all *p* < 0.001) (Table 3). Significant effects on CH_4_ emissions per kg milk were found for farm, week of lactation, parity, dry matter intake (all *p* < 0.001) and the interaction between dry matter intake and diet (*p* < 0.05).

There was no effect of diet on CH_4_ emission rate, but CH_4_ emissions per kg milk were higher for cows fed PMR with grazing (*p* < 0.001). Estimated mean CH_4_ emissions ranged from 0.2 g/min for cows at farms J and K to 0.6 g/min for cows at farms L and M (Figure 1a). Estimated mean CH_4_ emissions ranged from 15 g/kg milk for cows at farms J and K to 42 g/kg milk for cows at farms L and O (Figure 1b).

Rate of CH_4_ emissions increased to 0.4 g/min at week 9 of lactation, but were relatively constant from weeks 10 to 70 of lactation (Figure 2). Emissions of CH_4_ per kg milk generally increased to a peak in week 46 of lactation (Figure 2). Variability in emissions was more notable in later lactation. Furthermore, older cows were associated with a lower emission rate and per kg milk (both *p* < 0.001; Table 3). Increasing dry matter intake increased emission rate, with the increase being higher in grazing cows (0.02 g/min per kg dry matter intake). Increasing dry matter intake reduced emissions per kg milk, with the reduction being higher for cows fed PMR (−0.82 g/kg milk per kg dry matter intake).

When CH_4_ emissions had been adjusted for significant fixed effects, considerable residual variation in CH_4_ emissions remained among cows within farms. Coefficient of variation in CH_4_ emissions ranged from 5.7 to 75.1% for g/min and from 19.6 to 80.2% for g/kg milk for cows across farms.

Profile of CH_4_ emission rate throughout lactation was consistent among herds. Rank correlation for CH_4_ emission rate among weeks of lactation was generally high across lactations with 80% of rank correlations being greater than 0.5 (Figure 3).

Rank correlation for CH_4_ emission per kg milk among weeks of lactation were generally lower than for emission rate with 71% of rank correlations being greater than 0.5 (Figure 4).

## 4. Discussion

Although CH_4_ emissions increased overall with increasing dry matter intake, emission per kg milk decreased as dry matter intake increased. The increase in CH_4_ emission rate during early lactation reflects increased feed intake and milk production of cows during early lactation. Change in CH_4_ emission rate with week of lactation was similar to results reported in previous studies [4,6]. Diet composition and feed intake are important factors driving variation in enteric CH_4_ emissions from ruminant animals [13]. In the current study, there was considerable variability in enteric CH_4_ emissions among farms, however, ranking of cows for emission rate and emissions per kg milk were fairly consistent among herds. This supports implementation of farm-level spot measurements, which are repeatable at different stages of lactation. Farms with the lowest CH_4_ emission rate were Farms J and K, where cows were fed a PMR only. However, some farms, such as Farms A and B, fed cows a PMR with grazing and CH_4_ emissions were relatively low, which may be influenced by higher average age of cows in these herds. The farm with highest CH_4_ emissions was farm L, which may be due to farm L having heavier cows with higher feed intakes and milk production. In the current study, first and second-parity cows had higher emission rates than later parity cows. This may be due to the physiology of younger cows as they develop towards maturity, and as passage rate, digestibility and retention time of substrate in the rumen may be longer [14]. CH_4_ emissions in earlier parities were higher due to the digestive system of younger cows still developing [14,15].

Across farms, lower emissions per kg milk with a PMR diet than with PMR plus grazing may be due to a combination of higher proportion of concentrate feed in the diet reducing CH_4_ per unit of feed intake [16,17] and increased efficiency of energy utilisation from dilution of maintenance energy requirements. To meet their genetic potential for milk production, cows must maximise feed intake [18], which is more likely to be achieved with a highly digestible mixed ration than with pasture. In the current study, cows received a commercial concentrate feed during milking in addition to the concentrate component of PMR. Concentrate allowance fed during milking varied with individual milk yield. Because concentrate feed has a curvilinear effect on fibre digestion, a higher concentrate allowance would lower CH_4_ emissions per unit intake [19]. The higher emission rate in grazing cows can be explained by higher proportions of forage in diets of grazing cows (0.59) compared to those fed solely PMR (0.51).

The remarkable decline in CH_4_ intensity in dairy production during recent decades has been achieved through better nutrition and breeding [5,20]. Future technologies for further reducing CH_4_ emissions include genetic selection, feed additives [21,22] and on-farm CH_4_ monitoring. Frequent CH_4_ spot measurements from individual dairy cows during a day and over at least seven days after week 10 of lactation can provide a suitable estimate of individual animal emissions. Spot sampling of dairy cows whilst milking or feeding has been found to relate to total daily CH_4_ emissions [4,23,24,25] from the same cows when in a respiration chamber.

## 5. Conclusions

This study found significant effects of farm, week of lactation, parity, dry matter intake and the interaction between dry matter intake and diet, on CH_4_ emission rate and emissions per kg milk. Increasing dry matter intake increased emissions per minute, with the increase being higher in grazing cows. In addition, increasing dry matter intake reduced emissions per kg milk, with the reduction being higher for cows fed PMR. In terms of rank correlation, the profiles for CH_4_ emission rate and CH_4_ emission per kg milk during a lactation appear consistent among herds. Rank correlations for CH_4_ emissions among weeks of lactation was generally high across lactations.

## Figures and Tables

**Figure 1 animals-13-00157-f001:** Estimated mean CH_4_ emissions (with s.e. bars) in (**a**) grams per minute and (**b**) grams per kg milk for farms A to R.

**Figure 2 animals-13-00157-f002:** Estimated mean (with s.e. bars) CH_4_ emission rate (g/min; solid line) and per kg milk (dashed line) from week 1 to 70 of lactation.

**Figure 3 animals-13-00157-f003:** Rank correlation of CH_4_ emissions (with s.e. bars) in grams per minute across lactations.

**Figure 4 animals-13-00157-f004:** Rank correlation of CH_4_ emissions (with s.e. bars) in grams per kg milk across lactations.

**Table 1 animals-13-00157-t001:** Mean (s.e.) parity, milk yield, live weight, robot concentrate, dry matter intake (DMI) and number of milkings per day at each farm (A to R) for cows fed on diets consisting of a partial mixed ration (PMR), or PMR with grazing, plus concentrates.

Farm	Diet	No. of Cows	Parity	Milk Yield (kg/day)	Live Weight (kg)	Robot Concentrate DMI (kg/day)	DMI ^1^ (kg/day)	Milkings per Day (no. per Cow)
A	PMR + grazing	55	3.1 (0.2)	27.7 (1.2)	-	6.0 (0.4)	-	1.9 (0.07)
B	PMR	70	3.2 (0.3)	21.1 (0.8)	621 (7.8)	6.2 (0.2)	17.6 (0.2)	1.7 (0.05)
B	PMR + grazing	66	4.0 0.3)	21.8 (0.9)	597 (8.5)	4.6 (0.2)	17.1 (0.3)	1.9 (0.07)
C	PMR	41	2.1 (0.3)	30.2 (1.3)	634 (10.8)	9.2 (0.6)	18.9 (0.3)	1.9 (0.09)
C	PMR + grazing	34	2.9 (0.3)	24.7 (1.6)	634 (10.5)	7.3 (0.5)	19.3 (0.3)	2.0 (0.07)
D	PMR + grazing	47	2.2 (0.2)	26.9 (1.4)	610 (8.9)	7.0 (0.5)	17.9 (0.3)	2.0 (0.09)
E	PMR	73	2.6 (0.2)	26.4 (1.0)	647 (8.1)	7.3 (0.4)	18.8 (0.3)	2.1 (0.06)
E	PMR + grazing	71	3.9 (0.4)	28.2 (0.9)	629 (6.8)	7.3 (0.3)	18.5 (0.2)	2.4 (0.08)
F	PMR + grazing	45	3.6 (0.3)	26.3 (1.2)	601 (11.5)	5.0 (0.3)	17.6 (0.3)	2.4 (0.09)
G	PMR	116	2.6 (0.1)	25.5 (0.7)	627 (7.1)	5.9 (0.2)	18.2 (0.2)	2.3 (0.06)
H	PMR + grazing	85	3.0 (0.2)	26.2 (1.1)	-	7.9 (0.2)	-	3.4 (0.17)
I	PMR	110	2.9 (0.2)	25.8 (0.7)	602 (7.3)	5.1 (0.2)	17.6 (0.2)	2.1 (0.05)
J	PMR	329	2.7 (0.1)	31.0 (0.6)	669 (3.7)	6.6 (0.2)	19.9 (0.1)	2.3 (0.04)
K	PMR	199	2.2 (0.1)	28.8 (0.7)	-	5.9 (0.2)	-	2.4 (0.06)
L	PMR	63	3.7 (0.2)	27.0 (1.1)	697 (8.1)	5.3 (0.3)	20.1 (0.2)	2.9 (0.10)
M	PMR	119	2.4 (0.1)	34.3 (0.9)	611 (6.8)	6.4 (0.2)	18.7 (0.2)	2.3 (0.06)
N	PMR	129	2.0 (0.1)	22.4 (0.8)	605 (6.9)	6.6 (0.4)	17.4 (0.2)	2.6 (0.08)
O	PMR	81	2.9 (0.2)	18.7 (0.8)	580 (7.9)	5.7 (0.2)	16.4 (0.2)	2.5 (0.08)
P	PMR	26	2.4 (0.4)	29.7 (2.0)	-	4.0 (0.4)	-	2.2 (0.12)
Q	PMR	224	2.6 (0.1)	35.4 (0.9)	-	-	-	2.1 (0.03)
R	PMR	223	2.4 (0.1)	32.6 (0.6)	608 (4.8)	5.2 (0.1)	18.5 (0.1)	2.5 (0.06)
All		2206	2.6 (0.04)	30.4 (0.2)	617 (1.6)	5.7 (0.05)	18.5 (0.01)	2.4 (0.02)

^1^ Estimated by the equation: DMI (kg/day) = 0.025 × live weight (kg) + 0.1 × milk yield (kg/day) [12].

**Table 2 animals-13-00157-t002:** Forage and concentrate percentage and nutrient content in the dry matter (DM) of the diet for each farm (A to R).

Farm	Forage (% DM)	Concentrate (% DM)	DM (g/kg)	Starch (g/kg DM)	Neutral Detergent Fibre (g/kg DM)	Crude Protein (g/kg DM)	Oil (g/kg DM)	Metabolisable Energy (MJ/kg DM)
A	68.4	31.6	231	11	356	231	33	11.1
B	68.7	31.3	395	141	399	141	42	10.6
C	45.5	54.5	356	37	357	184	38	11.9
D	57.9	42.1	461	165	403	131	31	11.4
E	37.5	62.5	590	90	445	153	50	10.9
F	75.6	24.4	362	32	401	156	29	11.3
G	60.2	39.8	365	162	299	107	23	10.6
H	59.8	40.2	404	62	434	137	41	10.3
I	65.5	34.5	303	16	442	207	42	11.0
J	42.3	57.7	519	47	383	138	39	11.4
K	39.5	60.5	325	23	469	169	57	11.8
L	47.9	52.1	489	87	481	116	45	10.1
M	49.4	50.6	668	96	448	129	26	10.8
N	58.3	41.7	380	20	301	162	19	11.5
O	68.0	32.0	466	129	373	131	31	11.6
P	45.1	54.9	510	89	410	133	33	11.6
Q	42.9	57.1	360	19	470	155	50	11.2
R	56.0	44.0	361	68	411	131	49	10.5

**Table 3 animals-13-00157-t003:** Effect of farm, week of lactation, parity, diet and dry matter intake (DMI) on enteric methane emission rate (g/min) and per kg milk for dairy cows across commercial farms studied.

		Emissions (g/min)	Emissions (g/kg Milk)
Variable		Effect (s.e)	Mean	F-Statistic	d.f.	SED	*p* Value	Effect (s.e)	Mean	F-Statistic	d.f.	SED	*p* Value
Farm ^1^				129	17	0.02	<0.001			54	17	2.2	<0.001
Week of lactation ^2^				4.2	69	0.01	<0.001			10	69	1.4	<0.001
Parity	1		0.39	9.7	4	0.004	<0.001		26	16	4	0.48	<0.001
	2		0.39						26				
	3		0.38						25				
	4		0.37						25				
	5+		0.36						25				
Diet	PMR		0.38	0.8	1	0.007	0.385		24	22	1	0.78	<0.001
	PMR + grazing		0.38						28				
DMI		0.01 (0.001)		105	1	0.001	<0.001	−0.82 (0.08)		101	1	0.08	<0.001
Diet × DMI	PMR	0.01 (0.001)		17	1	0.002	<0.001	−0.82 (0.08)		5	1	0.32	0.024
	PMR + grazing	0.02 (0.003)						−0.10 (0.40)					

^1^ Farms A to R with estimated means shown in Figure 1. ^2^ Weeks 1 to 70 with estimated means shown in Figure 2.

## Data Availability

The datasets analysed are available from the corresponding author on request.

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
