# Peer review of "Variability in Enteric Methane Emissions among Dairy Cows during Lactation"

_animals, 2022, doi:10.3390/ani13010157_

Round 1

Reviewer 1 Report

This paper addresses an important and controversial subject regarding animal agriculture.  Overall, the paper is well written and easy to follow.  The authors use of the term 'predicted' added some confusion when I read through the paper.   To me this implies a prediction of an outcome of something that didn't happen yet.  Would the word 'estimate' be appropriate to use?  I believe that this is what the author's intend. 

The CH4 sampling in this study was done at a focal time point (at milking) when the cows were eating concentrate.  I would like to see a short discussion if the investigators feel that this would be representative of CH4 production at other times when they are not being milked or eating.  In other words, when they are relaxing and chewing their cud, would we expect CH4 production to be different?  If we are comparing data from this study to others that use more mobile analysis units that sample throughout the day, this may lead to discrepancies.  You do mention in lines 53/54 that this is a "reliable and repeatable measure" but does this compare well to other studies employing more continuous CH4 sampling?

Line

 Comment

26

Sentence is confusing as written.  Sounds like we directly increased the prediction of dry mater intake as an independent variable.   As long as this doesn't change the intended meaning,  I recommend  rewording to "Predicted dry matter intake was positively correlated with emissions rate, which was higher in grazing cows, and negatively correlated with emissions per kg milk and was most significant in PMR-fed cows."

125

Why are these referred to as 'predicted' emissions?  Did the authors decide to use this term in recognition that there is likely error in the CH4 calculation?   This leads to confusion on the part of the reader that we are making a prediction ahead of time what the CH4 emission will be.  I don't think that is what was done.  I would recommend using the word 'estimated'. 

138-147

The authors have used "Predicted mean CH4" and "CH4 emissions" throughout.  Is there a difference?  Please clarify.

Reviewer 2 Report

There are physiological mechanisms of ruminant animals to get rid of gases originated by rumen fermentation and being confined and demanded at a production level, methane production is potentiated and one of many strategies is through feeding. I suggest the review of the following publication: Animal Production Science, 2016,56, 1017–1034Reviewhttp://dx.doi.org/10.1071/AN15222

Reviewer 3 Report

The authors must consider only few details:

Please add more information about:

Grazing: grass species, spent time of grazing

Concentrates: cereal grains? commercial mix?

Table 2.- More details are required regarding:

Calculation of percentages of forage and concentrate in the diet (Line 86),

Laboratory techniques used for determination of crude protein, NDF, starch and oil,

Concentration of metabolysable energy (how did you estimate it?)

Statystical analysis:

Parity was a considered effect in the analysis (L118). In the used equation (model) it seems that this variable is named as “LN” (L120). I would assume that they are initials for “lactation number”. Please check and consider for avoiding misunderstandings.

Please add the description (meanings) of the components of the equation (below equation).
